# Understanding Compost-Bedded Pack Barn Systems in Regions with a Tropical Climate: A Review of the Current State of the Art

**DOI:** 10.3390/ani14121755

**Published:** 2024-06-10

**Authors:** Rafaella Resende Andrade, Ilda de Fátima Ferreira Tinôco, Flávio Alves Damasceno, Carlos Eduardo Alves Oliveira, Mariana Silva Concha, Ozana de Fátima Zacaroni, Gianluca Bambi, Matteo Barbari

**Affiliations:** 1Department of Biosystems Engineering, College of Agronomy, Federal University of Goiás, Goiânia 74690-900, GO, Brazil; 2Department of Agricultural Engineering, Federal University of Viçosa (UFV), Viçosa 36570-900, MG, Brazil; iftinoco@ufv.br (I.d.F.F.T.); carloseoliveira@ufv.br (C.E.A.O.); 3Department of Engineering, Federal University of Lavras (UFLA), Lavras 37200-900, MG, Brazil; flavio.damasceno@ufla.br; 4Department of Animal Science, School of Veterinary and Animal Science, Federal University of Goiás, Goiânia 74690-900, GO, Brazil; marianaconcha@discente.ufg.br (M.S.C.); ozacaroni@ufg.br (O.d.F.Z.); 5Department of Agriculture, Food, Environment and Forestry, University of Firenze, 50145 Firenze, Italy; gianluca.bambi@unifi.it (G.B.); matteo.barbari@unifi.it (M.B.)

**Keywords:** animal welfare, dairy cattle, housing systems, compost-bedded pack barns, heat stress

## Abstract

**Simple Summary:**

The search for housing systems that contribute to increases in productivity and milk quality with the meticulous use of resources is one of the main challenges with modern livestock. In this sense, the compost-bedded pack barn (CBP) is a promising alternative for raising dairy cattle. However, answers regarding the applicability of this system to the conditions of tropical and subtropical climates are lacking, focusing on Brazil. The objective of this study was to gather and describe the most recent information on open and closed CBP for dairy cattle. Properly designed, open CBP facilities with suitably designed ventilation systems and effective bedding management show potential for the climatic conditions and building typologies in Brazil. Most studies on the closed CBP system have provided only preliminary impressions because limited amounts of data have been collected. The first results demonstrate serious challenges with bedding management. The discussed results can be used to guide decision-making processes to create suitable environmental conditions for CBP systems.

**Abstract:**

The main challenge in milk production has been to maintain a focus on efficient processes that enhance production outcomes while aligning with animal welfare and sustainability and being valued by society. As an alternative to improve cow welfare in production and provide better handling of the waste generated by the activity, the system called the compost-bedded pack barn (CBP) has been widely adopted in countries with temperate climates and higher milk production. This CBP has been attracting global interest, including from countries with tropical and subtropical climates, such as Brazil, where many producers have started to use it due to the response in terms of milk productivity. A CBP can be designed either in (a) an open facility with natural ventilation or a positive-pressure ventilation system or (b) totally closed facilities, equipped with negative-pressure ventilation systems and permanent thermal control. The latter system is being implemented in Brazil, despite insufficient knowledge about its efficiency. The objective of this study was, through a review, to gather and describe the most recent information on the use of open and closed CBP systems for dairy cattle housing, mainly covering how it is applied in tropical climate regions. To achieve the proposed objective, this review study included the following topics related to CBPs: (i) implementation, (ii) bedding, (iii) general construction and architectural characteristics, and (iv) ambient thermal conditioning. Knowledge gaps and directions for future research are also identified here.

## 1. Introduction

One of the major concerns in the dairy farming sector worldwide is alleviating the negative effects of inadequate housing on cows during their lactation period. In addition to the adoption of modern technologies to improve the internal environment of livestock facilities, there is demand for facilities that ensure the sustainability of animal production, as highlighted by the trend of preservation of the environment and animal welfare that is currently signaled by consumers.

Simultaneously, there is an ongoing search for housing systems that reduce emissions, reuse waste, and ensure the efficiency of the return on invested capital [1]. Notably, the livestock sector represents an important source of emissions of ammonia and greenhouse gases and other impacts on the environment [2].

Another challenge in future housing for dairy cows is the creation of projects that resolve conflicts in existing systems, one of which is the amount of surface area required per animal [1]. More space per animal offers the possibility for more natural behavior, but tends to increase ammonia emissions per cow due to the larger emission surface per cow [3]. Additionally, when the lay public is introduced to the types of animal management systems, they think that living a natural life is an important part of animal welfare, reflecting their wishes for animals to live in natural environments with space and the ability to engage in species-specific behaviors [4].

When considering the aforementioned factors, the confinement of dairy cattle in compost-bedded pack barn (CBP) systems has shown promise for dairy farming. Such systems have been successfully used for several years with dairy cattle in temperate regions [5,6] and, recently, in tropical and subtropical countries such as Brazil [7,8].

The main reasons that have aroused the interest of dairy farmers in the CBP system include the increased comfort, health, and longevity of animals; improved waste management; and ease of completing daily tasks [5]. In this alternative animal husbandry system, cows remain in an extensive resting area, where they are offered a pasture-like environment in which they may lie down and stand up [9]. Producers use a conventional bedding system and incorporate composting methods through the periodic addition of carbon source material and daily turning of the bedding, promoting the composting of organic material [6,10].

Most CBP facilities are open on the sides and can be ventilated naturally or with mechanical positive-pressure ventilation [8]. Recently, some closed CBP facilities equipped with mechanical negative-pressure ventilation systems have been built in Brazil [11,12]. However, as the adoption of this milk-production-system technology has expanded in Brazil, concerns have also arisen from producers regarding the real applicability of totally closed facilities for the construction types and climatic conditions present in the country due to the limited research on this type of system. In view of the above, the objective of this review was to gather and describe the most recent information in the literature on open and closed CBP systems for dairy cattle housing, with the main focus being on how they are applied in regions with a tropical climate. The following topics related to the CBP system are discussed: (i) implementation, (ii) bedding, (iii) general construction and architectural characteristics, and (iv) ambient thermal conditioning. Knowledge gaps and directions for future research are also identified.

## 2. Methods

A literature search was conducted using the WoS and ScP databases, and all of the results returned in the searches were included in the Mendeley^®^ software, version 1.19.8, from which duplicates were identified and excluded. The search was not limited by the year of publication and all relevant papers published up to August 2023 were included. Only experimental articles written in English that had been peer-reviewed were considered. The criteria for the inclusion and exclusion of articles were defined a priori.

In order to carry out a more comprehensive study of the available literature in relation to the proposed theme, no additional restrictions, such as restrictions on the publication period, sample size, or quality of the journal, were imposed. Microsoft Excel^®^, version 2404, was used to extract and organize information of interest contained in the selected studies.

## 3. Compost-Bedded Pack Barn Systems

### 3.1. Implementation of Compost-Bedded Pack Barn Systems

#### 3.1.1. Globally

With the improvement in loose housing (LH) in the 1980s, the first CBP system reported in the literature was built by dairy farmers in Virginia, the USA. Later, in October 2001, in southern Minnesota, the USA, producers from that region had the idea of building a different facility with a new concept without re-producing any of the intensive milk production systems that had been previously adopted [6].

In Israel, the first CBP system was developed in 2006 and was quickly adopted among dairy farmers; no organic material other than cow feces and urine was added to the bedding area in the facility [13]. This method of operating a CBP system in this region was possible due to its arid conditions as the beds did not require additional material [14].

In the USA, the composting process within a CBP was improved by adding sawdust or wood chips as bedding material; this material was turned over two or three times per day, enabling aerobic degradation, which resulted in an increase in the internal temperature of the bed [14]. An increase in bed temperature is important for assisting in the composting process, that is, the aerobic degradation of organic material. 

As described, the concepts of the CBP systems used in the USA and Israel are quite different and have provided a basis for the development of other systems worldwide [1]. Thus, the CBP system have been disseminated to several regions, arousing the interest of several producers, especially in the United States, Canada, New Zealand, Germany, Italy, The Netherlands, Austria, Denmark, and South Korea [10,13,15,16].

Recently, CBP systems have started to be adopted in countries in South America, mainly in Brazil, Argentina, Colombia, Paraguay, and Uruguay, with variations in the characteristics and commonly used construction materials and ventilation systems [16,17,18,19,20]. However, few studies have been conducted regarding their applicability and efficiency for the specific climatic conditions of these countries as they have only recently been adopted in these regions.

There are differences in the models of CBP systems developed around the world. However, no standard solution is available for all agricultural, climatic, economic, and social contexts. CBP systems have the potential to increase animal welfare and longevity [1].

#### 3.1.2. Brazil

Milk production in Brazil, which was estimated at 35.30 billion L in 2021, is distributed across almost the entire country; the southeast, central-west, and south regions are most prominent—particularly Minas Gerais, Goiás, Paraná, Santa Catarina, and Rio Grande do Sul [21]. The average productivity worldwide is 2660 L of milk per cow while the Brazilian average is 2280 L per cow. The southern, southeastern, central-western, and northeastern micro-regions are well above the world average (4560 L per cow) and have productivity comparable to that of New Zealand, the largest dairy exporter in the world [22].

In order to ensure that Brazilian dairy farming maintains satisfactory production results, dedication and investment are increasingly demanded from producers to adapt their activities and overcome the numerous challenges posed by the country’s climatic conditions. Therefore, facilities for the confinement of dairy cows must be well designed to maximize animal comfort and mitigate the effects of climatic factors that may negatively impact production quality.

Facilities located in tropical and subtropical climates face different challenges from those in temperate countries as they must deal with both high temperatures and high relative humidity during much of the year [8,23]. The main difficulty experienced by milk producers in these regions is maintaining a high average monthly productivity throughout the year to make production more financially attractive. This has led producers to look for more adequate facilities to overcome the challenges arising from the weather conditions [9,12]. Dairy cows are sensitive to extreme temperatures. In tropical and subtropical regions of Brazil, thermal stress can reduce animal productivity. High levels of humidity, which are common in several parts of Brazil where milk production is high, such as in the south and southeast, can affect animal comfort. On the other hand, in the central-western region of Brazil, which has a hot and dry climate, the temperature can cause a high level of thermal stress for animals and harm food production.

For this purpose, the first open (conventional) CBP system was built in Brazil in 2012 and was located in the state of São Paulo [12]. The CBP system was implemented following the North American model, which was recommended as a low-cost and economically viable animal confinement method for Brazilian producers, as the building was open on the sides, favoring internal ventilation [1,12].

Although the conventional CBP system (with open sides) has been widely adopted by Brazilian dairy farmers, some producers have started to build closed and climate-controlled CBP systems. Through the implementation of closed CBP systems, producers seek to mitigate variations in the thermal environment between seasons and facilitate system management [8,12]. Figure 1 depicts images of the internal part of each type of system to detail the characteristics of open and closed CBP facilities.

The pressure ventilation in closed and climatized CBP systems is negative as a result of tunnel-ventilation, which is associated with evaporative cooling systems (ECSs). The first climatized closed CBP system was built in the state of Minas Gerais, Brazil, in 2015; since then, this method has been adopted in several regions of the country [12]. The main objective of a closed and climatized CBP system is to ensure more uniform ventilation and to provide better comfort inside the facility, especially during the hottest periods of the day. However, a lack of insulation of the side enclosures and roofs can drastically compromise the effectiveness of such a system.

Studies evaluating the thermal behavior of closed and climatized CBP facilities in Brazil are scarce [24]. Concomitantly in these systems, there are some undesirable points that are critical to the welfare of dairy cattle and have been internationally recognized as harmful to them. Excessive moisture in bedding can be harmful to the health of animals, and studies have observed a higher incidence of mastitis and hoof problems. Proper bedding management is the key point in this type of facility. During winter, a higher incidence of pneumonia has also been found in animals.

This type of housing system has only recently been scientifically tested as questions have been raised by producers, researchers, and technicians about both the management of such systems as well as the impacts on the environment and product quality [8,24]. The structure of closed CBP systems causes many problems, especially after the increased adoption of these systems by Brazilian producers, with numerous facilities being constructed without consulting a specialist/designer in the area [12].

Currently, the exact number of open and closed CBP systems in Brazil is unknown, but the number of new systems has rapidly grown [1,24]. Closed CBP systems already exist in the southeastern and southern regions of Brazil, with a few in the northeast [8,12]. The scientific data in the literature indicate that, to date, only closed CBP systems are found in Brazil.

Based on the above, to assess the applicability of open and closed CBP systems in Brazil, some relevant points that will be discussed here are the following: bedding management practices, the constituent material of the bedding, the bedding area per animal, the construction characteristics, and ventilation systems, among others.

### 3.2. Bedding for Compost-Bedded Pack Barn Systems

#### 3.2.1. Materials

When choosing the best bedding material, nutrients for microorganisms, animal comfort, availability, and cost-effectiveness should be considered [5]. In open and closed CBP systems for dairy cows, sawdust and shavings have been commonly used as bedding materials [1,7,8]. The use of carbon sources as bedding materials has been satisfactory, which is mainly due to their combination of absorbency and structural form, indicating their suitability for CBP systems [10].

However, in some regions in southeastern Brazil, a mixture of wood shavings and dry coffee husk has been used [12,25]. In the central-western region of Brazil, rice husks are predominantly used, yielding satisfactory results for producers due to the lower bedding replacement rate. The hot and dry climate of the region also makes it more favorable for using CBP systems and this type of bedding material. Radavalli et al. [26] observed that in the west of the state of Santa Catarina, Brazil, the most commonly used materials that produced adequate results as bedding in open CBP systems were 70% sawdust, 26.7% a mixture of sawdust and shavings, and only 3.3% wood shavings.

However, on many commercial dairy farms, bedding materials are selected based on their economic feasibility [27]. Additionally, the selection of bedding is dependent on its availability and cost at different times of the year. Attention must be paid to the type and management of the chosen bedding material as it is the main source of exposure of cows to the mastitis pathogen [16,28].

The success in the bedding component of open and closed CBP systems depends on the management of the composting process, the application of the material in the field, and its acquisition cost [1]. When materials are widely available, a system can substantially contribute to the globally discussed circular economy [2].

#### 3.2.2. Animal Bed Surface

The bedding area (m^2^∙cow^−1^) is one of the key aspects of the design of open and closed CBP systems. Smaller bedding areas per animal concentrate larger volumes of urine and feces, generating more moisture in the bedding and posing management difficulties. In a study carried out on open CBP systems in the south of the state of Minas Gerais, Brazil, 10.4 m^2^∙cow^−1^ was adopted as the bedding area [25]. An area from 11 to 19 m^2^∙cow^−1^ was used in the state of São Paulo [16], 14.6 m^2^∙cow^−1^ was used in the west of the state of Santa Catarina [26], and 16.4 m^2^∙cow^−1^ was used in Paraná [7]. In Brazil, for lactating cows, the minimum bedding area per animal is 10–14 m^2^∙cow^−1^ [24]. For closed CBP systems, the bedding area was 10 m^2^∙cow^−1^ in the Zona da Mata region of the state of Minas Gerais [12] and 10.5 m^2^∙cow^−1^ in the west of the state of Minas Gerais [8].

The differences observed in the bedding area per animal are due to several factors, such as the climate in each location, the construction of each facility, the ventilation system that was adopted, the ventilation rate for drying the bedding, the rate of turning the bedding, and the type of compost material. In hot, dry, well-ventilated climates, bedding is likely to dry faster, resulting in a reduced space that is available for each cow [10]. However, in cold and wet weather conditions, large amounts of material may be required to keep the surface adequately dry and comfortable for cows [1].

In CBP systems, Eckelkamp et al. [29] observed the influence of the environment on the bedding temperature, finding that the annual variations in the surface temperatures of the bedding in CBP facilities were similar to those of ambient temperatures. Likewise, Black et al. [15] evaluated CBP facilities and observed an average bed surface temperature of 10.5 ± 8.0 °C. According to the authors, evaporation and ventilation cool and dry the surface of the bedding in a CBP facility, causing the surface temperature level of the bed to be close to the ambient temperature. The larger the number of cows is and, consequently, the larger the amount of feces and urine is, the greater is the need to expand the bedding area per animal so that microbial activity and surface drying are balanced according to the daily amount of deposited manure [10].

#### 3.2.3. Management Practices

Producers must manage their facilities to keep bedding areas dry and, thus, avoid worsened cow hygiene and increased somatic cell counts [29]. For open CBP systems, the literature recommends turning the bed once, twice, or three times per day, preferably during periods when the animals are milking [1,10]. In both types of CBP systems (open and closed), the main objective of the turning process is to introduce oxygen into the upper layers of the bedding, promoting aerobic microbial degradation, which causes the heat produced from the process to help with drying the surface of the material [5,30]. Klaas et al. [13] emphasized that the generation of heat from the composting processes is crucial for the functioning of the system. Another important point is avoiding turning the compacted soil base together with the bedding material; this is more likely when the bedding depth is less than 0.30 m [10].

In a study conducted on a closed CBP system located in the state of Minas Gerais, Brazil, during summer and winter, bed turning was performed twice per day; however, the problem of excessive bedding moisture still occurred [12]. As the best strategy for the operation of this system with tunnel ventilation, under the specified conditions, a larger bedding area per animal and more frequent bed turning should be adopted [8,12].

The turning depth varies according to the management practices of the producer and the type of implement used; however, for open CBP facilities, studies recommend depths of 0.18 to 0.30 m [5,10]. In an open CBP in Rio Grande do Sul, Brazil, Silva et al. [31], found that the turning depth was influenced by the bed height. In most of the investigated facilities (60%), the turning depth was between 0.20 and 0.30 m.

The duration between additions of new layers of bedding material depends on the available area per animal, weather conditions, ventilation rate (air exchange), and type of bedding material. More bedding replacement is necessary during the rainy season and when there is insufficient air exchange inside the bedding [10,15,32]. Eckelkamp et al. [29] and Llonch et al. [33] indicated that more bedding material must be added when the bedding humidity is above 60%. The bedding management practices for closed and open CBP systems are the same, with more material being added when the bedding humidity is above 60% [12].

In open and closed CBP systems, specific management is required for both solid (bed area) and liquid (from cleaning the feeding corridor) waste [10,12]. Waste disposal must comply with legislation on the proper destination. Generally, solid and liquid residues are used as fertilizers in Brazil [8].

Typically, in open CBP systems located in temperate countries, the bedding is completely renewed every 6 to 12 months [1]. In countries with a tropical climate, the entire bed is renewed every 12 to 36 months. For closed CBP systems in Brazil, a shorter period between bedding replacements was observed, with an average period of 6 months [12]. This shorter time is probably related to the increased difficulty management beddings due to their high moisture content.

The renewal period of the bedding material depends on the construction characteristics of the facility, the available bedding area per animal, the ventilation systems adopted, the moisture content, and the management practices, among other factors. According to Klaas et al. [13], the compost removed from a facility can be directly spread on fields when necessary, and the transport costs are lower than those of liquid manure from other types of facilities. In open and closed CBP systems, additional benefits include the use of composted material to store animal waste and the possibility of marketing the material to generate additional income for the farm [32,34]. In traditional confinement systems for dairy cattle (free stalls and tie stalls), manure management represents a growing challenge for producers, requiring high investment in systems for the proper treatment of waste. CBP facilities with a supply corridor (with a concrete floor) generate approximately 30% less liquid effluent than traditional free-stall systems [35]. Subsequently, the generated liquid waste can be used to produce biofertilizers, thereby reducing the use of chemical fertilizers on fields, reducing expenses, and helping mitigate environmental issues.

The success of integrating bedding material into a system largely depends on the skills of those managing the composting process, the application of the material on the field, and the cost of acquiring the material [2]. Importantly, the incidence of mastitis in CBP systems is directly linked to the quality of the bedding. One of the indications that bedding is not of adequate quality is animals’ cleanliness score and the incidence of mastitis; clean animals experience less dirt adherence, reducing the probability of mastitis [30,34,36].

### 3.3. Construction and Architectural Characteristics in Brazil

The adequacy of CBP systems has been demonstrated by evidence of better comfort for dairy cattle [37,38]. Structural components have a substantial effect on cows’ microenvironments [39]. A wide variety of construction options exist for CBP systems; however, an ideal solution has not yet been identified. An extensive open resting area (bed) and the daily turning of the material seem to be the only common features of CBP systems worldwide [1].

Firstly, the producer needs to remember that the appropriate design of an animal building must consider all points of construction, including the choice of the site, the orientation of the building, earthworks, the definition of the foundation, and the budget and details of the material, in addition to electrical and hydraulic planning and finishing the concrete floor [40].

In general, the physical structure of open and closed CBP facilities is composed of a rest area (bed), food corridor, treatment track, feeders, drinkers, walls, and an access passage to the food corridor. In both types of CBP, cows have free access to the feeder, drinkers, and rest area [8,12]. Open CBP systems have sometimes been combined with access to pastures [6].

The first design criterion to be considered for open CBP facilities is the orientation of the structure. For Brazilian climatic conditions, they should be built with the longitudinal axis of the ridge oriented in the east–west direction [8]. This prevents direct solar radiation in the bedding area during the hottest hours of the day and can prevent the grouping of animals in certain areas, a factor that can compromise the quality of the bedding [16].

For closed CBP systems located in Brazil, regarding their orientation, in the west of the state of Minas Gerais, a northeast–southwest orientation was used [8]; in the region of Zona da Mata, Minas Gerais, a northwest–southeast orientation was used [12]. However, even for closed CBP facilities with non-insulating materials, an east–west orientation must be prioritized. In this way, during the hottest hours of the day, the solar radiation that enters the facility will be reduced.

The roof type varies among countries and depends on the climate, precipitation, wind speed, and snow load, among other factors [3]. In Brazil, to reduce construction costs, for open and closed CBP facilities, galvanized steel or aluminum tiling is commonly used; this type of material has a low absorption coefficient when new (high reflective power) and a high value of thermal conductivity, and, therefore, low insulating power [24].

The use of materials with increased thermal resistance for the roof allows an efficient increase in the control of the internal temperature of the facility [41]. In hot climates, these materials can reduce the heat flow from the roof to the facility, allowing for improved thermal comfort [42]. Low-quality materials and the general inadequacy of the structures lead to further difficulties in controlling the internal microclimate. A roof pitch between 15° and 25° is satisfactory for open CBP systems. The roof slope in closed CBP systems with climate control may also be lower than that of open systems because the air outlet is mechanically powered by exhaust fans [12]. In addition, closed CBP systems have a lining that helps with the thermal conditioning inside the facility (Figure 1b).

In open CBP systems, to prevent excess moisture from entering the bed, the roof eaves should not be less than 1.0 m in length [43]. Larger eaves prevent rain from entering and minimize solar radiation entering the interior of the facility. Oliveira et al. [25] observed the predominance of 2.0 to 3.0 m eaves in open CBP facilities in the south of the state of Minas Gerais, Brazil. For closed CBP installations, which have negative pressure ventilation, side closure with tarpaulins consequently prevents the ingress of rainwater, avoiding the need for wide eaves [30]. This finding was confirmed by Andrade et al. [12], who observed that the eaves were 0.8 m wide. In the same study, the side closure and ceiling lining of the facility (without insulation) were made of blue polypropylene. In addition to the characteristics of the materials of the side enclosures, other factors that influence the internal thermal environment of animal facilities, such as ventilation, the penetration of solar radiation, and processes or pieces of equipment that release heat inside the building, must be considered [43].

In Brazil, an open CBP facility width of approximately 20.0 m has been often used [24,25]. In the south of the state of Minas Gerais, Brazil, a facility length of 73.3 m is common [25]. In the case of a closed CBP facility located in the State of Minas Gerais, Andrade et al. [12] observed a width and length of 26.4 m and 55.0 m, respectively. To date, no consensus has been reached on the ideal size of a bed in closed CBP facilities; however, to reduce the cost of earthworks and the difficulties that are the management of the animals and bedding, facilities with longer than 200 m should be avoided [44].

As observed by Oliveira et al. [25] and Radavelli [26], the average ceiling height for open CBP systems in Brazil is taller than 4.3 m. In the case of a closed CBP system, Andrade et al. [24] observed that the ceiling height was 4 m. The ceiling height of a closed CBP system must be shorter to reduce the volume of internal air. A taller ceiling requires a larger number of exhaust fans. However, opening the system later will be difficult due to the lower ceiling height of the facility. 

When deciding to build a shed with a controlled environment, the height of the ceiling should be lower than that of open sheds, facilitating the control of the internal temperature by the cooling system. However, for open and closed CBP systems, the selection of the height should consider the execution of routine work, such as the entrance of wagons and tractors, which requires a height of approximately 4.0 m [45].

Equally important when planning the facility is defining whether structures such as the milking parlor, waiting room, and waste pit will be attached to the cow housing shed [40]. This decision will affect the positioning of the corridors needed for the animals to move to these structures. In addition, the dimensioning of these structures must be linked to the objective of the final project so that the facilities do not have to be readjusted in the future, generating undue expenses.

The floor level under the bed area must be designed to keep the bed surface level with the floor of the feed aisle, with the bed depth varying from 0.20 m to more than 1.0 m. Depending on the country and legislation, the floor under the bed may or may not be paved [1].

In open and closed CBP systems, the bed area is normally separated from the feed aisle by a short wall, which is usually 1.2 m high [28]. To prevent the bedding material from leaving the facility, in open CBP systems, a masonry wall with a height between 0.3 m and 0.5 m, which is associated with a steel cord fence, has commonly been implemented to guarantee the proper circulation of indoor air [25]. This wall can also prevent rain ingress and the bed from spreading out of the facility. In addition, in Brazil, lower wall heights have been adopted to favor natural ventilation. In closed CBP systems, side closures prevents excess bedding from spreading out of the facility and the rain from entering.

The length of the feeder also strongly influences the design as it is related to the length of the facility. In open and closed CBP systems, feeders should be easily accessible to animals, and a linear feeder space of 0.7 to 1.4 m per cow is recommended [43]. 

The length of the handling lane (the aisle where the food is offered) must be the same as the length of the feeding aisle, and the width must be sufficient for machines (such as tractors) to be able to work. Generally, widths of 4.0 m are sufficient. The food aisle must be at least 4.0 m wide [10]. Another recommendation is for the entire area of the feed aisle and treatment lane to be covered in concrete. A CBP system potentially requires less concrete than an FS system. In open and closed CBP systems, the supply aisle can be positioned in the facility on only one side, laterally on both sides, or centrally [30].

In most open and closed CBP facilities, drinking fountains are located along the walls that divide the feeding aisle and bedding area, facing the feeding aisle [24,43]. Radavelli et al. [26] investigated the construction of 30 open CBP systems located in western Santa Catarina, Brazil. The results showed that some of the open CBP systems contained drinking fountains in the bedding area or nearby so that the animals could drink while in the bedding area. Drinkers in this region are not recommended as they can add more moisture to the bedding and negatively influence the composting process. In a study on closed CBP systems in the state of Minas Gerais, Brazil, the drinking fountains were located on the walls that divided the bed area from the food corridor and facing the food corridor [12].

Another important point for both system types is cleaning the floor of the food aisle, which must be cleaned daily. The feeding corridor, which has a concrete floor, collects 25 to 30% of the manure and urine produced by the animals [6].

Attention should also be paid to the lighting provided in open and closed CBP systems, mainly to facilitate the operators’ work and the inspection of animals, as well as to assist in handling tasks. A well-lit facility can increase animal movement, improve food intake, and, consequently, increase milk production [12].

In places where continuous general lighting is used or areas with simple visual tasks (for example, in the bed region), the illuminance must be 100 to 200 lx. For the feeder region, a minimum light level of 200 lx is recommended [35,42]. Andrade et al. [35] evaluated the spatial distribution of luminosity in a closed CBP. According to the study results, the luminosity values were below the recommended values (with an average of 84.96 lx, varying between 3.0 and 492.67 lx) for dairy cows during their lactation period. Closing the sides of the facility influenced the luminosity intensity, indicating that the intensity and distribution of luminosity needed to be changed.

The other facilities that are essential for the production process in open and closed CBP systems include the milking parlor, waiting room, calf stall, bull stall (when artificial insemination is not used), chute or trunk for vaccination and spraying, silos, feed deposit, and forage chopper compartment [40]. Another important aspect is determining whether structures such as the milking room, waiting room, and waste pit will be attached to the cow housing shed during the planning phase. This decision will affect the placement of corridors to take the animals to these structures [30].

The milking parlor, which is normally automated, must be connected to the confinement facility so that two or three milkings can be performed daily under hygienic conditions. Some properties that have closed CBP systems have opted to place the milking parlor and waiting room inside the facility next to the evaporative plates. As such, the animals remain in a controlled environment throughout the lactation period.

### 3.4. Effects of the Environment on the Thermal Comfort and Welfare of Dairy Cows

Facilities that provide more space per cow and soft bedding and allow free movement have been proven to be beneficial for animal comfort and well-being [46]. The proper ventilation of a system can reduce the impact of heat stress on animals. Fávero et al. [16] suggested managing CBP system bedding to provide a dry, loose surface, resulting in cleaner animals with a lower incidence of mastitis. According to the same authors, the hygiene score of cows can be a useful and efficient indicator to help manage bedding and assess the risk of subclinical mastitis in CBP systems.

According to Blanco-Penedo et al. [46], the hygiene of the animal’s body and udder reflects the cleanliness of that surface where the animal lies. In a study conducted on open CBP and free stall (FS) facilities, hygiene levels were assessed on a five-point scale (1 = clean and 5 = very dirty) [36]. The mean hygiene scores for open CBP facilities were better than those of FS facilities, averaging 1.95 ± 0.09 versus 2.18 ± 0.06, respectively. However, Eckelkamp et al. [29] did not observe differences between the average hygiene scores of cows in CBP (2.19 ± 0.05) and FS (2.26 ± 0.06) facilities, which indicated that the animals in both systems remained clean throughout the study.

Marcondes et al. [47] evaluated the productive characteristics of dairy farms located in the state of Minas Gerais, Brazil, that switched from a confinement system (dry-lot, DLS) to open CBP facilities and compared them with similar farms that did not change their rearing system. The authors observed that farms with open CBP facilities increased their milk production per cow by 13.3% compared with farms with DLS confinement and concluded that these results were probably due to the better environmental conditions and greater animal comfort provided by open CBP facilities.

Burgstaller et al. [48] noted that open CBP facilities were a good alternative to FS facilities in terms of lameness, hoof health and animal welfare. Bran et al. [18] conducted a study on intensive farms in southern Brazil to investigate factors associated with lameness in dairy cows. The same authors observed that farms that used mattresses as a base had a higher prevalence of lameness than farms that used composted bedding.

### 3.5. Ambient Thermal Conditioning Systems Used in Compost-Bedded Pack Barn Facilities

The thermal environment is a composite of a series of factors that characterize the microclimate inside facilities; these factors interact these factors each other and reflect the real thermal sensation of the animals [12]. Concomitantly, the appropriate thermal environment can optimize the feed efficiency of the animals [46]. However, facilities without adequate control of microclimatic variables may struggle to provide thermal comfort, and heat stress conditions can affect animal productivity. Under these conditions, animals use mechanisms to regulate their body temperature, which can lead to a high level of stress, resulting in reduced well-being and further reductions in milk production [49]. Although the effect of cold stress on milk production is minimal, the effect of heat stress on milk production can be extremely harmful [47,48].

Adequate ventilation in both open and closed CBP systems is necessary to remove excessive heat and humidity from the ambient air, as well as the heat and humidity generated by the bedding, and to ensure the hygienic quality of the ambient air [1,50]. The ventilation process varies according to the design of the facility adopted for dairy cows in Brazil. The two main categories of CBP systems are as follows:(a)CBP systems designed in open facilities in which internal air renewal naturally occurs through natural ventilation or is supplemented with positive-pressure ventilation (Figure 2a).(b)CBP systems designed in closed facilities, where the sides are closed, and both the volume of internal air as well as the volume and flow of air that enters and leaves are controlled. The ventilation is achieved through a negative-pressure system (Figure 2b).

**Figure 2 animals-14-01755-f002:**
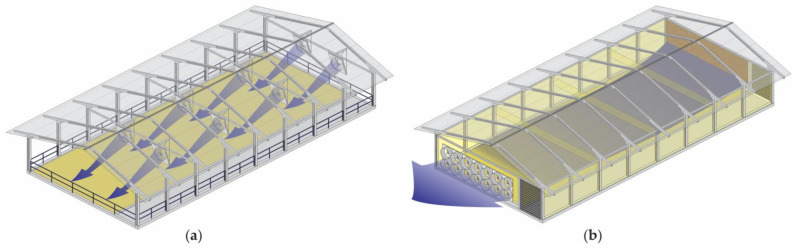
Compost-bedded pack barn systems with (**a**) open sides and positive-pressure ventilation and (**b**) closed sides and negative-pressure ventilation associated with an adiabatic evaporative cooling system. Source: the authors.

#### 3.5.1. Positive-Pressure Ventilation Systems

When natural ventilation is insufficient to improve the indoor environment of a facility, fans over the bed area and the food aisle must be used to reduce the ambient heat, improve the air quality, and promote bed drying [45]. According to Leso et al. [1], the main types of fans found in open CBP systems are high-volume and low-speed (HVLS) or low-volume and high-speed (LVHS) fans. It should be mentioned that in Brazil, most CBP facilities are open and use positive-pressure ventilation with LVHS fans [8,12,25].

Oliveira et al. [25] analyzed twenty CBP facilities located in the south of Minas Gerais, Brazil, and they observed a predominance of facilities with LVHS fans (76.4%). The mean values of air velocity at the bed surface and at a height of 1.5 m were 1.3 ± 0.7 m∙s^−1^ and 1.7 ± 0.8 m∙s^−1^, respectively. According to Black et al. [15], an air velocity of approximately 1.8 m∙s^−1^ close to the surface of the bedding provides a higher rate of drying of the bedding. In addition, maintaining adequate wind velocity in the cows’ resting area is important to reduce the physiological responses associated with heat stress such as increased respiration rates [39]. 

According to Damasceno et al. [43], ventilation must be homogeneous so that animals are not crowded in a specific area of the facility. A lack of fans can cause animals to crowd in areas where the natural airflow is higher during heat stress conditions, which leads to the accumulation of manure and urine. In this situation, composting is inefficient due to the increase in moisture content.

The adequate dimensioning of the ventilation system depends on the types and models of fans available and should be based on their rotation, power, flow, and performance, in addition to the air speed required in the environment, and it should be supported by the animals without stress (which depends on the quantity and size of the animals). This adequate dimensioning also depends on the placement and size of the air inlets and outlets. The automation involved in controlling the activation of specific components—including the automated handling of fans—must be considered (Figure 3). Fans that operate at variable speeds—normally controlled by thermo-hygrometers—should be chosen as they can work in places where the temperature and relative humidity of the external air vary widely during the day [8,12]. The operating speed of these fan models can be changed as a function of thermal comfort indices such as the temperature–humidity index (THI) and enthalpy (H).

A combination of fans to increase convection heat loss and sprinklers/nebulizers to promote adiabatic evaporative air cooling is effective in cooling dairy farming environments [27]. Sprinkler systems can be installed either in the waiting room or in the feeding corridor of a CBP system as they make up one of the most efficient methods of reducing thermal stress in animals [30].

In Brazil, installing the necessary number of fans in facilities is difficult due to a lack of electricity that can occur in some regions in the countryside. In addition, in some areas, if producers have a generator with insufficient demand or if they have constant power outages, they choose to use solar energy or a biodigester (Figure 4).

It is important to highlight that sprinklers should not be installed in the bed area. Compost bedding requires an optimal moisture content to facilitate the microbial activity necessary for composting [5]. Adequate moisture helps maintain the biological processes that break down organic matter. Excessive moisture can create anaerobic conditions, which inhibit the composting process and lead to the production of undesirable odors and potentially harmful gases such as ammonia [1]. Excessive water can lead to the leaching of nutrients and potential contamination of groundwater [44].

Overly wet bedding can also become compacted, reducing cow comfort and increasing the risk of hoof problems and other health issues [10]. Wet bedding can lead to an increased incidence of mastitis and other infections due to the proliferation of bacteria in moist environments [51]. Excessive moisture can lead to foot problems such as lameness, significantly impacting animal welfare and productivity [1].

#### 3.5.2. Negative-Pressure Ventilation Systems

Since the early 2010s, new confinement systems have gained popularity in dairy farming worldwide. The use of closed system in association with evaporative cooling has been adopted in free-stall facilities with tunnels or cross-ventilation [45].

The adoption of this type of climate control has produced substantial advantages for dairy production by drastically reducing exposure to heat stress and improving animal comfort in comparison with open confinement or pasture systems [18,48]. 

Fully enclosed facilities require mechanical ventilation and evaporative cooling systems to allow reductions in air temperature during the hottest months of the year and the hottest hours of the day [8,12,52]. A climatized system is based on the control, direction, and cooling of the air inside a facility used for animal production.

As reported by Fournel et al. [27] and Mondaca et al. [39], systems with negative-pressure ventilation are usually automated. In tunnel-type systems, the air is sucked through exhaust fans along the length of the installation (Figure 5a); in systems with cross-ventilation, the air is sucked perpendicularly to the length of the building (Figure 5b). In both cases, the herd is exposed to winds at practically constant speeds. The ventilation design typically follows some assumptions regarding the unit of flow per animal, air exchange for a specified period, and air velocity in the cross-sectional area [39]. A properly designed negative-pressure ventilation system can provide uniform air movement throughout a facility.

In such a system, in hot weather conditions during the driest periods, the hot and unsaturated air that is external to the facility is forced (by the exhaust fans positioned at the opposite end) to cross a moistened porous plate, resulting in a simultaneous exchange of heat and mass. This leads to a change in state of part of the water from the liquid phase to the vapor phase and an increase in the relative humidity of the inlet air, with a consequent reduction in temperature. That is, the temperature of the indoor air decreases but the air has an increase in its relative humidity [35]. For regions with hotter and drier climates, the use of this evaporative pad cooling system can decrease the temperature of the air by up to 11.0 °C. 

Vega et al. [53] reported a first approach to modeling and simulating the thermal distribution inside a tunnel-ventilated CBP facility located in a tropical environment with an evaporative cooling system using computational fluid dynamics (CFD). They observed that the average difference in the dry-bulb temperature between the external and internal conditions was from 2.1 to 5.8 °C. They recommended increasing the airflow velocity to above 3 m·s^−1^ when the external dry-bulb temperature and the relative humidity simultaneously exceed 30 °C and 55%, respectively.

Air tends to move through aisles, ceilings, or the feed aisle, where interference with cows is minimal, i.e., the path of least resistance [54]. The set of exhausters must be dimensioned to provide a minimum air speed ranging between 2.0 and 4.0 m·s^−1^. For systems with tunnel ventilation, a ventilation rate of 1700 m^3^·h^−1^ per cow and an inlet air velocity of at least 2.5 m·s^−1^ should be adopted [40].

The capacity of the hoods must be adequate to guarantee the necessary air renewal rates in the summer. The time required for all of the hoods to completely renew the indoor air of the facility must be calculated, and a complete air change in one minute or less is desirable [40]. An air change is the equivalent of replacing all of the air inside the building with fresh air, i.e., if the air change rate is one minute, every minute, the exhaust fans move enough air to completely replace the air inside of the facility with outside air [8,12,55]. However, as the ventilation rate increases, the operational energy costs for running the hoods also increase.

A negative-pressure system that is associated with adiabatic evaporative cooling operates through an automatically controlled panel that activates exhaust fans and the wetting of porous plates. In such a system, sensors that monitor the environmental conditions are positioned inside buildings, and they allow the cooling system to be activated; normally, when the air temperature is equal to or greater than 21 °C and the relative humidity of the air reaches values close to 80%, the cooling system is turned off [40,56].

Controllers can be used to adjust the inlet opening, fan rotation, and trigger for wetting the porous material based on the indoor air temperature and relative humidity. They must be installed close to the centers of the ventilated areas and at least three points along the length of the facility [35].

Maintaining a well-programmed handling routine is important for the proper functioning of a system because, whenever the gates are opened, hot air enters the building, causing peaks in internal air temperature. Peripheral air entry—for example, due to sealing failures—should also be avoided as it can cause a reduction in system efficiency. Advanced planning, careful observation, and corrections after the negative-pressure system is installed can minimize performance issues. The most common problem is the presence of dead air spaces that do not have sufficient air velocity or fresh air [39].

However, the structural components of facilities can affect the performance of a ventilation system as they can considerably reduce its efficiency and cost-effectiveness [39]. Moreover, the failure to appropriately dimension the air inlet dimensions can interfere with the pressure of the system, leading to pressure drops, the overloading of the exhaust fans, and, consequently, an increase in electrical energy consumption. Some dairy farmers who have adopted this type of system have chosen to place the milking parlor inside a closed CBP facility so that the animals are exposed to a climatized environment throughout the period.

This type of climatized system also requires continuous mechanical ventilation and the use of a quality generator that is properly designed to maintain operation in the event of a power failure [40,45,52]. Dairy producers should consider incorporating generators into their projects. Such systems are highly dependent on electrical energy; even with the opening of curtains, if the electrical energy is not quickly restored, the internal air temperature can remarkably increase, causing heat stress in animals [24]. This also increases the concentration of ammonia and other pollutants in the air, creating an unhealthy environment for animals and workers.

The longitudinal or transversal airflow provided by this type of climatized environment carries the metabolic heat produced by the confined animals, the composting of the bedding, the thermal load generated by the equipment used, and the solar radiation emitted through the roof and side enclosures. It also promotes the transport of air pollutants (high concentrations of heat, humidity, ammonia, dust, etc.). This results in an increase in the air temperature, as well as a worsening of the air quality near the hoods.

According to Damasceno et al. [44], narrower installations provide more uniform air, and buildings with a width of more than 20.0 m require additional air inlets that must be inserted close to the food aisle. According to the same author, no common consensus has been reached on the ideal length for a closed CBP system. However, in practice, buildings up to 180.0 m in length have been constructed, requiring further studies to determine the ideal length [8,24]. Andrade et al. [12] observed that even in installations with a length of 55.0 m, excessive bedding moisture was a problem; this was probably related to the high animal density, excessive relative humidity, inefficiency of the ventilation system, and need for more frequent turning of the bedding. In this type of closed facility, the addition of a ceiling lining is recommended to reduce the cross-sectional area and the volume of internal air, thus increasing the system efficiency and lowering the cost of implementation, as the ceiling height usually varies between 2.5 and 4.0 m [35]. Attention must also be paid to the height of the lining so that it does not interfere with the entry of tractors and trucks into the facility. The ceiling acts as a second physical barrier that allows the formation of a mobile air layer next to the roof, which contributes to the reduction in heat transfer to the interior of the building. A wider shed or one with a higher roof will require a larger fan capacity to maintain the air velocity over the cows [12].

Deflectors made of metal or canvas are also commonly used. These deflectors mainly have the function of redirecting the airflow to the location of the cows and increasing the air velocity, thus minimizing their heat stress [54]. Additionally, deflectors can be located to divert the airflow through dead air spaces.

Harner et al. [55] recommended that the bottom of the baffles should be high enough (3.6 to 4.0 m) to avoid interference with the normal operation of equipment. In the specific case of closed CBP systems, the deflectors must be installed at a height that does not interfere with the turning, replacement, and removal of the entire bed. In a closed CBP system with tunnel ventilation, the deflectors are installed in the transverse direction of the installation. In addition, placing a transverse deflector over the feeding aisle (arranged in the continuation of the deflectors that are located above the bed) and located longitudinally in the line of the feeders is common.

Deflectors must be strategically located to direct air into cow locations [55]. The design and proper location of the deflectors are essential to minimize the static pressure encountered by the exhaust fans because, as the static pressure increases, the performance of the exhaust fans decreases [52]. Figure 6 illustrates a schematic model of a closed CBP system with tunnel ventilation in association with evaporative cooling.

Several negative-pressure facilities have been incorrectly dimensioned when considering that the exhaust fans work at nominal flow; the load losses associated with animals, deflectors, and characteristics of the evaporative plate have often been ignored in calculations [24]. According to Tyson et al. [57], critical steps in the development of a negative-pressure ventilation system include (a) determining the capacity of the exhaust fans, (b) selecting the exhaust fans, (c) determining the size of the input, (d) installing fans and controls, and (e) selecting the locations of the fans and inlet.

#### 3.5.3. Evaporative Pad Cooling Process 

Evaporative cooling pads are usually composed of fibrous materials and are commonly added to the air inlets of tunnel-ventilated facilities [27]. The materials most used in the filling of evaporative coolers are fiberglass, cellulose, polypropylene, and wood fiber [35]. Cellulose has been preferentially used in the manufacturing of evaporative cooling pads, and it is predominant in animal production facilities in Brazil.

To prevent reductions in the cooling efficiency, the system must be serviced after a certain period of use, especially before and after summer, when the triggering happens more frequently. Evaporative plates may show signs of deterioration and obstruction due to exposure to too much dust. Plate clogging is also common, and it can be seen as flaws in the water path. Board materials deteriorate over time and must be replaced after a certain use period [58].

The operation of such systems is based on vertically or horizontally wetting plates of a porous material so that the air is forced through exhausters to pass perpendicularly through this material. The porous material can be wetted by dripping water on the upper edge of a vertically arranged plate or by spraying water on its surface [59].

The working principle of evaporative cooling is that the system can only remove sensible heat from the environment; therefore, it works best in hot and dry climates, where it produces maximum evaporative cooling [59,60]. In this climate-controlled animal husbandry system, the prior cooling of the air by forcing the passage of external air through plates with moistened porous material must be performed with caution. Although the system allows for a substantial reduction in air temperature, the relative humidity consequently increases, which may reduce the amount of heat dissipated by animals in evaporative form, creating moderate heat stress in animals in these internal environmental conditions [8,12].

An increase in the relative humidity of the air can hinder the drying of bedding, becoming a limiting factor that creates challenges for handling; negatively influences animal hygiene, milk quality, and the bedding composting process; increases the emission of harmful gases such as ammonia; and accelerates the oxidation of metallic parts, among other issues. Harner et al. [55] observed that the gases emitted by free-stall facilities with negative-pressure ventilation were predominantly nitrogen-based (ammonia, nitrogen dioxide, and nitric oxide) during study periods in the spring and summer seasons. In addition, an excessive amount of humidity inside a closed CBP system can cause early equipment wear (corrosion).

The capacity to lower the air temperature through an evaporative cooling system depends on the ambient temperature and the relative humidity of the air to be cooled. As the relative humidity increases, the capacity for air temperature reduction decreases [52]. Systems that cool the air through evaporative cooling are most effective in hot climates with low relative humidity; however, they can also be used in regions with high humidity during the hottest hours of the day, when relative humidity tends to naturally decrease. However, the effectiveness of these evaporative cooling systems is questionable in environments with a permanently humid climate [60].

As Brazil is a large country, the microclimate of each region must be carefully assessed to assist in decision making regarding the best facility design [30]. Closed CBP systems have potential for use in some regions of Brazil and can be used with higher efficiency depending on the climatic conditions. The northeastern, central-western, and southeastern regions have the potential to adopt this type of system; however, for the northern and southern regions—especially the coastal regions—increased caution is needed due to the high relative humidity [8,12,24].

Currently, there are few guidelines for the proper operation of closed CBP systems. However, the adoption of this type of system has markedly increased in recent years. Evaporative pad cooling and nebulizers should be used with caution in CBP systems—especially over the bedding area—because they can generate an increase in relative humidity, causing a decrease in bedding evaporation rates [1].

#### 3.5.4. Importance of Air Quality Inside Compost-Bedded Pack Barn Facilities

Inadequate ventilation can lead to the accumulation of gases, increasing the risk of health hazards for workers and animals. Effective ventilation systems are crucial to ensuring that harmful gases are continually removed from a facility’s interior environment. As mentioned by Rong et al. [61], livestock facilities are an important source of emissions of ammonia (NH_3_), methane (CH_4_), nitrous oxide (N_2_O), and carbon dioxide (CO_2_). Ammonia is responsible for eutrophication and soil acidification while CO_2_, CH_4_, and N_2_O have been identified as greenhouse gases that contribute to global warming [62].

In this sense, the contaminating gases produced during a CBP system’s bedding tilling process, especially during the hottest periods of the year, must be considered, as they are a significant factor in emissions derived from systems with composted bedding [63]. For this reason, despite the potential benefits for animal health and welfare, the contaminating gases from manure in composting systems must be considered.

The composting process that occurs daily in manure from barns with compost bedding leads to large amounts of CH_4_ and NH_3_ emissions [61]. A bed with excess moisture becomes more compacted, contributing to a greater accumulation of ammonia inside the installation [64]. Bewley et al. [65] reported that the smell of ammonia inside a CBP facility was noticeable when the carbon/nitrogen ratio was below 25:1, hindering the process of microbial activity and heat production in the bed.

Ammonia evaporation is related to the intensity of the evaporation of moisture from manure; a strong correlation between these parameters has been established. Ammonia evaporates up to 3.9 times from liquid manure more intensely than it does from solid manure [66]. The ammonia concentration in naturally ventilated buildings is mainly influenced by the air temperature. Therefore, to ensure good air quality in stables and reduce the pollution of air with harmful gases, it is important to properly manage the intensity of ventilation. As ventilation is intensified, ammonia evaporates more [66].

In closed installations with tunnel ventilation, the continuous monitoring of gases and adequate sizing and operation of exhaust fans must be required. In the event of ventilation failures, air quality is quickly impaired.

## 4. Conclusions and Future Perspectives

CBP systems are applicable in Brazil; however, adaptations of the North American model have been made to better suit the climate. A great challenge remains in maintaining the bedding at adequate temperature and humidity levels.

For open CBP systems in Brazil, mechanical ventilation, combined with a sprinkler system in the feeding aisle, is necessary. Natural ventilation is insufficient to alleviate thermal stress in animals. Therefore, it is recommended to choose a ceiling height greater than 4 m and open sides with low walls around the bedding area. Galvanized steel tiles are commonly used for this type of installation. Providing more than 10 m^2^ per cow in the bedding area has resulted in better bedding quality and good production outcomes. The bedding in a CBP system must be turned two to three times per day. The most commonly used bedding materials that have shown positive results in terms of durability and less frequent replacement are sawdust, wood shavings, and rice husks. Some properties use wood shavings together with coffee husks. The compost resulting from the bedding is used as organic fertilizer for crops, promoting the sustainability of the property. The installation of solar panels or other renewable energy systems should be considered to reduce the operating costs and environmental impact. However, for regions with very humid climates, the use of this type of system should be cautiously evaluated.

The installation model with closed sides and negative-pressure ventilation has been a major challenge for Brazilian producers. Bedding material replacement is performed more frequently than bed management in CBP systems with open sides and positive-pressure ventilation.

In the decision-making process regarding which type of CBP system to adopt, producers must consider several aspects that guide the planning of the production process—especially animal welfare, sustainability, and cost-effectiveness.

Previous research on closed CBP systems has provided preliminary impressions because only small amounts of data have been collected. As closed CBP systems are already a commercial reality in Brazil, research aimed at improving the construction of buildings, the ventilation systems used, and bedding management is required.

The dairy industry would also considerably benefit from further research on CBP systems as this information will refine approaches to the circular economy for the improvement of efficiency while reducing impacts on the climate and environment. Sustainable alternative systems for dairy cattle production are increasingly accepted by key stakeholders such as producers, specialists, and consumers.

## Figures and Tables

**Figure 1 animals-14-01755-f001:**
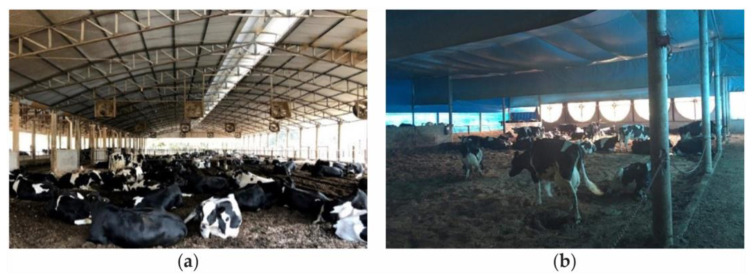
Interior of (**a**) open and (**b**) closed compost-bedded pack barn systems. Source: the authors.

**Figure 3 animals-14-01755-f003:**
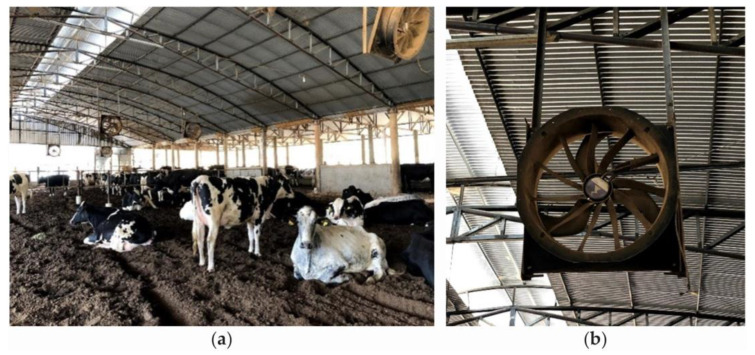
Compost-bedded pack barn system with open sides with automated fans: (**a**) distribution of fans and (**b**) detail of the equipment. Source: the authors.

**Figure 4 animals-14-01755-f004:**
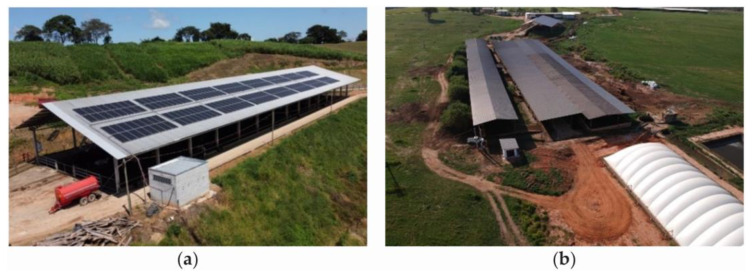
Open compost-bedded pack barn systems with low-volume and high-speed (LVHS) fans and (**a**) solar panels and (**b**) biodigesters. Source: the authors.

**Figure 5 animals-14-01755-f005:**
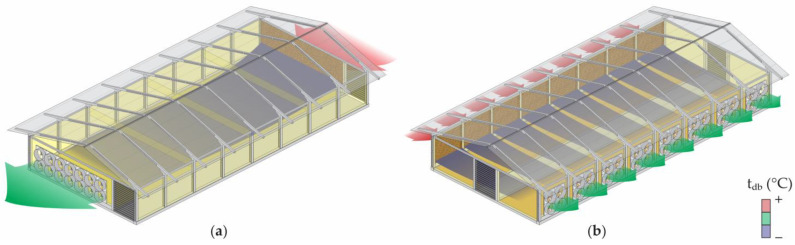
Mechanical exhaust ventilation system (negative pressure) in a compost-bedded pack barn system with (**a**) tunnel ventilation and (**b**) cross-ventilation. t_db_—dry-bulb temperature. Source: the authors.

**Figure 6 animals-14-01755-f006:**
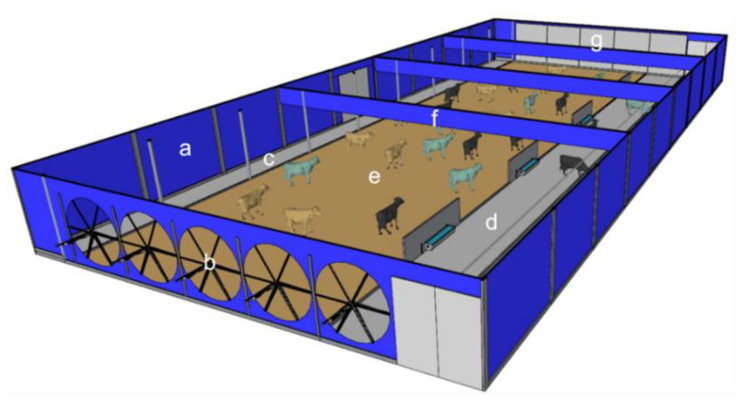
Schematic representation of an existing closed compost-bedded pack barn system with negative pressure ventilation, associated with evaporative cooling. a—side closing of curtains; b—high volume exhaust fans; c—service corridor; d—food corridor; e—rest area (bed area); f—deflectors; and g—evaporative cooling pads. Source: the authors.

## Data Availability

The data set is available on request to the corresponding author.

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
