# Peer review of "Understanding Compost-Bedded Pack Barn Systems in Regions with a Tropical Climate: A Review of the Current State of the Art"

_animals, 2024, doi:10.3390/ani14121755_

Round 1

Reviewer 1 Report

Comments and Suggestions for Authors

The manuscript develops a review about a topical aspect of farm building desig, i.e. the Compost-Bedded Pack Barn (CBP) Systems. Moreover, the review focuses on the Brazilian context and the tropical regions in general, which are lacking specific insighths in that field.

The manuscript is very well organized and correctly written; furthermore the main issues related to CBP were addressed in the text.

Therefore, only a few minor revisions are requested, as follows.

Lines 119-124: Mostly repetition of lines 110-114;

Lines 525-527: The effects of water on the compost bedding should be discussed;

Lines 592-628: The issue of air quality inside a closed CBP barn should be discussed more broadly, with particular reference to the gases produced by the composting process, their permanence in the barn indoor volume, as well as the effectivenbess of extraction fans to remove noxious gases.

Comments on the Quality of English Language

The English quality of the manuscript is good, and only minor revisions should be carried out by performing a careful re-reading, also considering the comments by reviewers.

Author Response

Dear Reviewer,

Thank you for your important contributions to improving this article. We have completed the English review. The article was corrected in English, and the Word proofreading tool was used to show them.

According to the English revision carried out, the title will now look like this: Understanding Compost-Bedded Pack Barn Systems in Regions with a Tropical Climate: A Review of the Current State-of-the-Art

Below are the modifications and explanations of the revision suggestions. The changes are highlighted in yellow in the text.

In the box if you only upload an attachment.

--

Lines 119-124: Mostly repetition of lines 110-114

Ok, lines 119-124 were removed.

Lines 525-527: The effects of water on the compost bedding should be discussed

Sprinklers should not be installed in the bed area. Compost bedding requires an optimal moisture content to facilitate the microbial activity necessary for composting (Barberg et al., 2007). Adequate moisture helps maintain the biological processes that break down organic matter. Excessive moisture can create anaerobic conditions, which inhibit the composting process and lead to the production of undesirable odors and potentially harmful gases like ammonia (Leso et al., 2020). Excessive water can lead to leaching of nutrients and potential contamination of groundwater (Damasceno, 2020).

Overly wet bedding can also become compacted, reducing cow comfort and increasing the risk of hoof problems and other health issues (Shane et al., 2010). Wet bedding can lead to increased incidences of mastitis and other infections due to the proliferation of bacteria in moist environments (Eckelkamp et al., 2012). Excessive moisture can lead to foot problems such as lameness, significantly impacting animal welfare and productivity (Leso et al., 2020).

Lines 592-628: The issue of air quality inside a closed CBP barn should be discussed more broadly, with particular reference to the gases produced by the composting process, their permanence in the barn indoor volume, as well as the effectivenbess of extraction fans to remove noxious gases.

3.5.4 Importance of air quality inside Compost-Bedded Pack Barn facilities

Inadequate ventilation can lead to the accumulation of gases, increasing the risk of health hazards for workers and animals. Effective ventilation systems are crucial to ensuring that harmful gases are continually removed from a facility's interior environment. As mentioned by Rong et al. [58], livestock facilities are an important source of emissions of ammonia (NH3), methane (CH4), nitrous oxide (N2O), and carbon dioxide (CO2). Ammonia is responsible for eutrophication and soil acidification, while CO2, CH4, and N2O have been identified as greenhouse gases that contribute to global warming [59].

In this sense, the contaminating gases produced during a CBP system's bedding till-ing process, especially during the hottest periods of the year, must be considered, as they are a significant factor in emissions derived from systems with composted bedding [60]. For this reason, despite the potential benefits for animal health and welfare, the contaminating gases from manure in composting systems must be considered.

The composting process that occurs daily in manure from barns with compost bedding leads to large amounts of CH4 and NH3 emissions [58]. A bed with excess moisture becomes more compacted, contributing to a greater accumulation of ammonia inside the installation [61]. Bewley et al. [62] reported that the smell of ammonia inside a CBP facility was noticeable when the carbon/nitrogen ratio was below 25:1, hindering the process of microbial activity and heat production in the bed.

Ammonia evaporation is related to the intensity of the evaporation of moisture from manure; a strong correlation between these parameters has been established. Ammonia evaporates up to 3.9 times from liquid manure more intensely than it does from solid manure [63]. The ammonia concentration in naturally ventilated buildings is mainly influ-enced by the air temperature. Therefore, to ensure good air quality in stables and reduce the pollution of air with harmful gases, it is important to properly manage the intensity of ventilation. As ventilation is intensified, ammonia evaporates more [63].

In closed installations with tunnel ventilation, the continuous monitoring of gases and adequate sizing and operation of exhaust fans must be required. In the event of ventilation failures, air quality is quickly impaired.

Reviewer 2 Report

Comments and Suggestions for Authors

This review article the existing literature covering Compost-Bedded Pack Barn Systems. It gives a summary of how CBP was developed, how it can be used, and what management is needed for its success. There is good use of photos and diagrams to illustrate the points. Some more detail is needed in places - see specific comments below. Overall, this is quite a long review, and could be slightly shortened through re-wording and consolidating the text throughout to improve readability.

The title does not quite make grammatical sense in its current form: 'A Review of the Current State-of-the-Art' either has word missing or needs re-phrasing in order to make sense e.g. a review of the current state of the art techniques, or something similar.

Line 17 - it is unclear why you mention breeding systems when your focus is on bedding systems.

Line 31-33: repetition of the word alternative

Line 49 and 54 - reference to breeding again - unclear what you mean here. Is this a typing/ spelling error?

Line 85 - repetition of review.

Line 134 - the last sentence should be integrated into the previous one.

Line 139 - how does it improve soil quality?

Line 142 - please give an example of common environmental parameters experienced in Brazil for reader context.

Line 149 - the start of this sentence does not make sense - the project conventional

Line 162 - what is tunnel mode? 

Line 171 - what welfare points specifically?

 Line 174 - breeding??

Line 180 - would be good to know a bit of background to Brazilian dairy industry in the introduction. Roughly how many cows and how many dairies in Brazil? Type of cows and average yield?

Line 194 - the meaning of this sentence is unclear - what does the 'use of materials' mean?

Section 3.2.1 - so what is the best bedding material to use? are there specific problems or benefits to any you mention?

Line 213 - is this total area, so including lying, loafing and feeding space?How do these measurements compare to other countries using CBP?

Line 240 - please give more detail on why turning the bed is needed. What mechanical equipment is needed to do this? You have given some in line 252, so please move this to the start of this section.

Line 294 - are there specific mastitis pathogens associated with CBP? What re common mastitis rates found in these systems?

Line 299 - how is cow comfort actually measured?

Line 308 - 's'?

Line 320 - why do closed systems need different orientation?

Line 338 - what lining?

Line 341 - are the large eaves to help let air into the shed?

Line 357 - why 200m?

Line 375 to 377- should this be in the earlier section on bedding area per cow?

Line 392 - that is a very large range of feed pace recommendations. can you justify what is best to use?

Line 438 - again, how is cow comfort measured?

Line 458 - please give some examples of lameness levels seen? what types of lameness, is digital dermatitis a problem?

Line 466 - how does it impact feed efficiency?

Line 526 - what is adiabatic cooling?

Line 584 - calculated

Line 631 - which author?

Line 682 - serviced how often?

Conclusion - overall you have reviewed a lot of material, but there are no firm conclusions drawn. After reading this review, i still do not know how i should build or manage a CBP system in Brazil to get the best outcomes. This must be addressed. Is your only conclusion that there isn't enough research? If a client asks you to design a CBP system, what will you pick based on the research reviewed? What are the key features it must have etc.

Comments on the Quality of English Language

Some formatting of English is needed throughout. Spelling of bedding and breeding needs checking throughout.

Author Response

Dear Reviewer,

Thank you for your important contributions to improving this article. The article was corrected in English, and the Word proofreading tool was used to show them.

Below are the modifications and explanations of the revision suggestions. The changes are highlighted in yellow in the text.

--

The title does not quite make grammatical sense in its current form: 'A Review of the Current State-of-the-Art' either has word missing or needs re-phrasing in order to make sense e.g. a review of the current state of the art techniques, or something similar.

A review of the Current State-of-the-Art is commonly used in the literature and can be seen in several articles, including Animals.

e.g. Huijsmans, T. E., Hassan, H. A., Smits, K., & Van Soom, A. (2023). Postmortem Collection of Gametes for the Conservation of Endangered Mammals: A Review of the Current State-of-the-Art. Animals, 13(8), 1360. https://doi.org/10.3390/ani13081360

According to the English revision carried out, the title will now look like this: Understanding Compost-Bedded Pack Barn Systems in Regions with a Tropical Climate: A Review of the Current State-of-the-Art

Line 17 - it is unclear why you mention breeding systems when your focus is on bedding systems.

replaced for: “housing systems”

Line 31-33: repetition of the word alternative

As an alternative to improve cow welfare in production and provide better handling of the waste generated by the activity, the system called compost-bedded pack barn (CBP) has been widely adopted in countries with temperate climates and higher milk production.

Line 49 and 54 - reference to breeding again - unclear what you mean here. Is this a typing/ spelling error?

Replaced for: One of the major concerns in the dairy farming sector worldwide is to alleviate the negative effects of inadequate housing environments on cows during their lactation period. In addition to the adoption of modern technologies to improve the internal environment of livestock facilities, a demand exists for facilities that ensure the sustainability of animal production, highlighted by the trend of preservation of the environment and animal welfare currently signaled by consumers.

Line 85 - repetition of review.

It's ok, it's been removed.

Line 134 - the last sentence should be integrated into the previous one.

Recently, CBP systems have started to be adopted in countries in South America, mainly in Brazil, Argentina, Colombia, Paraguay, and Uruguay, with variations in the characteristics and commonly used construction materials and ventilation systems [18–22]. However, few studies have been conducted regarding its applicability and efficiency for the specific climatic conditions of these countries, as it is a system that has recently been adopted in these regions.

Line 139 - how does it improve soil quality?

Ok, it's been removed.

Line 142 - please give an example of common environmental parameters experienced in Brazil for reader context.

Facilities located in tropical and subtropical climates face different challenges than those in temperate countries, as they must deal with both high temperature and relative humidity during much of the year [8,23]. The main difficulty experienced by milk producers in these regions is maintaining a monthly high average productivity throughout the year to make production more financially attractive. This has led producers to look for more adequate facilities to overcome the challenges arising mainly from the weather conditions [9,13]. Dairy cows are sensitive to extreme temperatures. In tropical and subtropical regions of Brazil, thermal stress can reduce animal productivity. High levels of humidity, common in several parts of Brazil where milk production is high, such as in the south and southeast, can affect animal comfort. On the other hand, in the central-western region of Brazil with a hot and dry climate, they can generate a high level of thermal stress for animals, as well as harm food production.

Line 149 - the start of this sentence does not make sense - the project conventional

We change to: The CBP system was implemented following the project of the North American model, being recommended, therefore, as a low-cost and economically viable animal confinement method for Brazilian producers, as the building is open on the sides, favoring internal ventilation [1,13].

Line 162 - what is tunnel mode? 

Tunnel-ventilated

Line 171 - what welfare points specifically?

Excessive moisture in the bedding can be harmful to the health of animals, studies have observed a higher incidence of mastitis and hoof problems. Proper bedding management is the key point in this type of facility. During winter, a higher incidence of pneumonia was also found in animals.

Line 174 - breeding??

it's been removed.

Line 194 - the meaning of this sentence is unclear - what does the 'use of materials' mean?

The use of bedding material, as carbon sources, has been satisfactory, mainly due to the combination of absorbency and structural form, indicating their suitability for CBP systems [25].

Section 3.2.1 - so what is the best bedding material to use? are there specific problems or benefits to any you mention?

When choosing the best bedding material, nutrients for microorganisms, animal comfort, availability, and cost-effectiveness should be considered [5]. In open and closed CBP systems for dairy cows, sawdust and shavings have been commonly used as bedding material [1,7,8]. The use of bedding material, as carbon sources, has been satisfactory, mainly due to the combination of absorbency and structural form, indicating their suitability for CBP systems [25].

However, in some regions in southeastern Brazil, a mixture of wood shavings and dry coffee husk has been used [13,26,27]. In the central-west region of Brazil, rice husks are predominantly used, with satisfactory results for producers due to the lower litter re-placement rate, the hot and dry climate of the region also makes it more favorable for the use of the CBP system and this type of litter material. Radavalli et al. [28] observed that in the west of the state of Santa Catarina, Brazil, the most-used materials producing adequate results as bedding in open CBPs were 70% sawdust, 26.7% a mixture of sawdust and shavings, and only 3.3% wood shavings.

Line 213 - is this total area, so including lying, loafing and feeding space? How do these measurements compare to other countries using CBP?

No, as described it is only the bedding area per animal.

Line 240 - please give more detail on why turning the bed is needed. What mechanical equipment is needed to do this? You have given some in line 252, so please move this to the start of this section.

Ok.

Line 308 - 's'?

Ok, it's been removed.

Line 320 - why do closed systems need different orientation?

We change to: For closed CBP located in Brazil, in relation to the orientation of the systems, in the west of the state of Minas Gerais, northeast–southwest orientation it was used[8]; in the region of Zona da Mata, Minas Gerais, northwest–southeast orientation was used [13]. However, even for closed CBP facilities with non-insulating materials, the east-west orientation must be prioritized. This way, during the hottest hours of the day, the solar radiation that enters the facility will be reduced.

Line 338 - what lining?

In closed installations it is common to use linings (Figure 1b).

Line 341 - are the large eaves to help let air into the shed?

No, larger eaves prevent rain from entering and minimize solar radiation entering the interior of the facility.

Line 357 - why 200m?

Reference added.

Line 375 to 377- should this be in the earlier section on bedding area per cow?

Yes, it has been moved to the section on bedding area per cow

Line 438 - again, how is cow comfort measured?

Facilities that provide more space per cow, soft bedding and allow free movement to have been beneficial for animal comfort and well-being [52]. Proper ventilation of the sys-tem can reduce the impact of heat stress on animals. Fávero et al. [18] suggest that managing the CBP system bedding to provide dry, loose surface results in cleaner animals with a lower incidence of mastitis. According to the same authors, the cow's hygiene score can be a useful and efficient indicator to help manage bedding and assess the risk of subclinical mastitis in CBP systems.

Line 526 - what is adiabatic cooling?

Evaporative Pad Cooling System

For regions with hotter and drier climates, the use of this evaporative pad cooling system can allow the e of the air by up to 11.0 °C [45,58].

Line 584 – calculated

Thanks!

Line 631 - which author?

Damasceno (2020)

Line 682 - serviced how often?

To prevent reductions in the cooling efficiency, the system must be serviced after a certain period of use, especially before and after summer, when the triggering happens more frequently.

Conclusion - overall you have reviewed a lot of material, but there are no firm conclusions drawn. After reading this review, i still do not know how i should build or manage a CBP system in Brazil to get the best outcomes. This must be addressed. Is your only conclusion that there isn't enough research? If a client asks you to design a CBP system, what will you pick based on the research reviewed? What are the key features it must have etc.

CPB systems are applicable in Brazil; however, adaptations of the North American model have been made to better suit the climate. A great challenge remains in maintaining the bedding at adequate temperature and humidity levels.

For open CPB systems in Brazil, mechanical ventilation, combined with a sprinkler system in the feeding aisle is necessary. Natural ventilation is insufficient to alleviate thermal stress in animals. Therefore, it is recommended to choose a ceiling height greater than 4 meters and open sides, with low walls around the bedding area. Galvanized steel tiles are commonly used for this type of installation. Providing more than 10 m² per cow in the bedding area has resulted in better bedding quality and good production outcomes. The bedding in a CPB system must be turned two to three times per day. The most commonly used bedding materials, that have shown positive results in terms of durability and less frequent replacement, are sawdust, wood shavings, and rice husks. Some properties use wood shavings together with coffee husks. The compost resulting from the bedding is used as organic fertilizer for crops, promoting the sustainability of the property. The installation of solar panels or other renewable energy systems should be considered to reduce the operating costs and environmental impact. However, for regions with very humid climates, the use of this type of system should be cautiously evaluated.

The installation model with closed sides and negative-pressure ventilation has been a major challenge for Brazilian producers. Bedding material replacement is performed more frequently than bed management in CPB systems with open sides and positive-pressure ventilation.

In the decision-making process regarding which type of CPB system to adopt, producers must consider several aspects that guide the planning of the production process—especially animal welfare, sustainability, and cost-effectiveness.

Previous research on closed CPB systems provides preliminary impressions because only small amounts of data have been collected. As closed CPB systems are already a commercial reality in Brazil, research aimed at improving the construction of buildings, the ventilation systems used, and bedding management is required.

The dairy industry would also considerably benefit from further research on CPB systems, as this information will refine approaches to the circular economy for the improvement of efficiency while reducing impacts on the climate and environment. Sustainable alternative systems for dairy cattle production are increasingly accepted by key stakeholders, such as producers, specialists, and consumers.

Round 2

Reviewer 2 Report

Comments and Suggestions for Authors

The script is much improved - thank you to the authors

Comments on the Quality of English Language

much better